# Flavonoids in *Amomum tsaoko* Crevost et Lemarie Ameliorate Loperamide-Induced Constipation in Mice by Regulating Gut Microbiota and Related Metabolites

**DOI:** 10.3390/ijms24087191

**Published:** 2023-04-13

**Authors:** Yifan Hu, Xiaoyu Gao, Yan Zhao, Shuangfeng Liu, Kailian Luo, Xiang Fu, Jiayi Li, Jun Sheng, Yang Tian, Yuanhong Fan

**Affiliations:** 1College of Agronomy and Biotechnology, Yunnan Agricultural University, Kunming 650500, China; 2Engineering Research Center of Development and Utilization of Food and Drug Homologous Resources, Ministry of Education, Yunnan Agricultural University, Kunming 650500, China; shengj@ynau.edu.cn; 3College of Food Science and Technology, Yunnan Agricultural University, Kunming 650500, China; 4Yunnan Key Laboratory of Precision Nutrition and Personalized Food Manufacturing, Yunnan Agricultural University, Kunming 650500, China; 5Department of Science and Technology, Yunnan Agricultural University, Kunming 650500, China; 6Yunnan Plateau Characteristic Agricultural Industry Research Institute, Yunnan Agricultural University, Kunming 650500, China; 7Yunnan Aromatic Bioengineering Technology Research Center, Yunnan Agricultural University, Kunming 650500, China

**Keywords:** *Amomum tsaoko*, constipation, gut microbiota, gut metabolites, gastrointestinal motility

## Abstract

*Amomum tsaoko* (AT) is a dietary botanical with laxative properties; however, the active ingredients and mechanisms are still unclear. The active fraction of AT aqueous extract (ATAE) for promoting defecation in slow transit constipation mice is the ethanol-soluble part (ATES). The total flavonoids of ATES (ATTF) were the main active component. ATTF significantly increased the abundance of *Lactobacillus* and *Bacillus* and reduced the dominant commensals, such as Lachnospiraceae, thereby changing the gut microbiota structure and composition. Meanwhile, ATTF changed the gut metabolites mainly enriched in pathways such as the serotonergic synapse. In addition, ATTF increased the serum serotonin (5-HT) content and mRNA expression of 5-hydroxytryptamine receptor 2A (*5-HT_2A_*), Phospholipase A2 (*PLA2*), and Cyclooxygenase-2 (*COX2*), which are involved in the serotonergic synaptic pathway. ATTF increased Transient receptor potential A1 (*TRPA1*), which promotes the release of 5-HT, and Myosin light chain 3(*MLC3*), which promotes smooth muscle motility. Notably, we established a network between gut microbiota, gut metabolites, and host parameters. The dominant gut microbiota *Lactobacillus* and *Bacillus*, prostaglandin J2 (PGJ2) and laxative phenotypes showed the most significant associations. The above results suggest that ATTF may relieve constipation by regulating the gut microbiota and serotonergic synaptic pathway and has great potential for laxative drug development in the future.

## 1. Introduction

Constipation is a common gastrointestinal disorder and can be divided into secondary constipation and functional constipation. Constipation is characterized by difficulty in passing stools or a low frequency of bowel movements, often accompanied by straining during defecation or a feeling of incomplete evacuation [1]. Slow transit constipation (STC) is the most common form of functional constipation. The pathogenesis of STC is mainly related to degenerative changes in the enteric nervous system (ENS), abnormal release of neurotransmitters, smooth muscle dysfunction, and an imbalance of the gut microbiota [2]. At this stage, the application of medications for constipation is still the predominant treatment and can be divided into herbal, Western, and microbial drugs, but the common drugs are still laxatives and prokinetics, which have all been found to have varying degrees of adverse effects in clinical application [1]. Therefore, it is of great clinical value and urgency to investigate a safe, effective and inexpensive gastrointestinal stimulant based on edible natural plant material for the treatment of chronic functional constipation [3]. *Amomum tsaoko* Crevost et Lemarie (AT) is a classic spice commonly used to make soups in Asian countries such as China, Thailand, Burma, and Vietnam. It is also a traditional Chinese herb, and decoctions of AT are commonly used by Chinese medicine practitioners to treat conditions such as cold dampness and internal obstruction, distension, and pain in the abdomen [4]. Recent studies have shown that the main chemical constituents of AT include volatile oils and nonvolatile components such as phenols, flavonoids, diphenylheptane and bicyclonononanes; AT has pharmacological effects such as regulating gastrointestinal functions [5] as well as anti-inflammatory [6], antitumor [7], hypoglycemic, and lipid-regulating effects [8]. Clinically, AT is commonly used in the treatment of gastrointestinal conditions, as in the use of AT aqueous extract (ATAE) to accelerate the recovery of gastrointestinal functions and the relief of abdominal distension in abdominal surgery patients [9]; thus, it is presumed that AT may have laxative effects. 

A previous study performed in our lab revealed that ATAE was effective in relieving symptoms of loperamide-induced constipation in mice [10]. However, the exact laxative ingredients are still unknown. Here, we used a murine model of loperamide-induced constipation to explore the key active component of ATAE in relieving constipation, and systematically further explored the laxative mechanism of the key active component of ATAE from the perspective of gut microbiota and intestinal metabolism in a mouse model of loperamide-induced constipation.

## 2. Results

### 2.1. Flavonoid-Rich ATES Is the Active Fraction Involved in ATAE Laxative Activity

Animal experiments 1 were to investigate the effects of ATAE on loperamide-induced constipation in mice (Figure 1A). In animal experiments 1, all mice appeared healthy with no abnormal behavior. Loperamide, Maren pill and ATAE treatment had no significant effect on body weight, food intake or water consumption (Appendix A). 

Compared to Control group (CON), the defecation time of the first black stool(FBST) of Model group (LOP) was significantly increased, while ATAE treatment reversed the time increase in LOP (Figure 1B), and the medium-dose group (MAE) of ATAE was the most effective and outperformed the positive drug. Fecal wet weight (FW) and Fecal number(FN) also confirmed the laxative effect of MAE (Appendix A), but no significant difference was observed in the fecal water content between groups (Appendix A).

To explore the main active fraction of ATAE, ATAE was isolated by ethanol precipitation to obtain the Ethanol precipitation of ATAE (ATEP) and the Ethanol-soluble part of ATAE (ATES), and the laxative effects of ATES and ATEP were evaluated in a mouse model of constipation (Figure 1C,D). In animal experiments 2 and 3, all mice appeared healthy with no abnormal behavior. Loperamide and ATES, ATEP treatment had no significant effect on body weight, food intake or water consumption (Appendix A). 

Compared with CON, the FBST of LOP was significantly increased, the treatment of ATES significantly reduced the time to FBST, and the effect of the medium-dose group of ATES (MES) was similar to that of MAE (Figure 1E). However, there was no significant difference in FBST between the three dose groups of ATEP (LEP, MEP, HEP) and LOP (Figure 1F). FW and FN verified the above results (Appendix A). Therefore, we determined that the key active ingredient is the ATES component. In addition, there was still no significant difference in fecal water content between groups in animal experiments 2 and 3 (Appendix A).

The phytochemical composition of ATES was examined using an extensive targeted metabolomics approach (HPLC-QQQ-MS/MS) (Figure 1G). The results showed that flavonoids accounted for 46.63% of the total composition, while the content of all other components was low. Based on this finding, we speculate that flavonoids may be the key active ingredients in AT laxation. Using porous polymer bead technology, the total flavonoids in ATES (ATTF) was purified from ATES (Figure 1H, Appendix A). In summary, ATAE and its key active component ATES have a good laxative effect, and after purification, ATTF in the ATES was increased to 72.22%.

### 2.2. ATTF Alleviated Loperamide-Induced STC Symptoms

Animal experiments 4 were to investigate the effects of ATTF on loperamide-induced constipation in mice (Figure 2A). In animal experiments 4, all mice appeared healthy with no abnormal behavior. Loperamide and ATTF treatment had no significant effect on body weight, food intake or water consumption (Appendix A).

Porous polymer bead technology was used to purify flavonoid-rich ATTF. The high dose of ATTF (HAF) significantly reversed the significant increase in FBST induced by loperamide (Figure 2B) and the significant decrease in FN and FW (Figure 2C,D). However, the fecal water contents were not significantly affected by HAF treatment (Figure 2E). In the gastrointestinal transit test, Gastrointestinal transit rate (GTR) was significantly reduced in LOP mice compared to CON mice, but HAF intervention had no effect on constipated mice (Figure 2G,J). Interestingly, loperamide caused a significant decrease in liver and kidney indices, while HAF administration reversed these changes to different extents (Figure 2H,I). Through pathological section observation, we found that the colonic mucosal barrier is slightly disrupted by loperamide, manifested by a discontinuity of colonic epithelial structures (indicated by arrows) and a slight increase in immune infiltration. ATTF reduced the destruction of the colonic mucosal structure caused by loperamide (Figure 2K). In summary, as the main component of ATES, ATTF is indeed effective in relieving constipation and reversing side effects that may be caused by loperamide.

### 2.3. ATTF Partially Restored Loperamide-Induced Gut Microbial Dysbiosis

Changes in the structure and composition of the gut microbiota play an important role in the progression of STC. The V3-V4 region of the 16S rRNA gene was sequenced to determine the effect of ATTF on STC model mice. The rarefaction curves for all the cecal content samples collected from the CON, LOP and HAF group mice tended to approach the saturation plateau (Figure 3A), suggesting that the sequencing depth was sufficient to cover the entire spectrum of bacterial diversity and was suitable for the present study.

In the α-diversity analysis, LOP and HAF were slightly lower than CON (Figure 3B); CON and LOP were clearly separated in the PCoA plot of β-diversity analysis, with HAF between them (Figure 3C). At the phylum, family, and genus levels, the community structural composition was significantly influenced by HAF (Appendix A), leading to the classification of CON and LOP into one category in the clustering tree using differential gut microbiota (Appendix A). Differential taxa of intergroups showed that HAF had some restorative effect on the gut microbiota of loperamide-induced STC mice.

In addition, we analyzed the abundance of gut microbiota that differed significantly between the LOP and HAF groups (Appendix A). The dominant microbiota at different levels for each group was obtained by using linear discriminant analysis (LDA) effect size (LEfSe) analyses (Appendix A). We focused on the taxa that were significantly affected by HAF treatment.

At the phylum level, the relative abundances of *Bacteroidota*, *Proteobacteria,* and *Campylobacterota* were reduced significantly compared to their levels in LOP (Figure 3E). At the family level, the relative abundances of 10 families were affected by the HAF intervention, either increasing or decreasing (Figure 3F–H). The relative abundances of the dominant families *Lactobacillaceae* and *Bacillaceae* (belonging to Firmicutes) increased to 54.2% and 11.4% in the HAF group, respectively (Figure 3F). Other families of Firmicutes decreased significantly with HAF intervention, mainly *Lachnospiraceae, Erysipelatoclostridiaceae, Oscillospiraceae, Eubacterium_coprostanoligenes_group, UCG-010* and *Anaerovoracaceae* (Figure 3G). In addition, *Muribaculaceae* of *Bacteroidota* and *Helicobacteraceae* of *Campylobacterota* decreased significantly with HAF intervention (Figure 3H).

At the genus level (Figure 3I–K), the relative abundances of the dominant genera *Lactobacillus* and *Bacillus* in the HAF group were significantly higher than those in the LOP group (Figure 3I). With HAF intervention, the relative abundances of six genera of Firmicutes significantly decreased, including *unclassified_f_Lachnospiraceae*, *Lachnospiraceae_NK4A136_group*, *Erysipelatoclostridium*, *unclassified_f_Oscillospiraceae*, *Anaerotruncus,* and *Family_XIII_UCG-001* (Figure 3J). In addition, the relative abundances of *Helicobacter* and *Odoribacter* were reduced by HAF treatment (Figure 3K). In general, HAF may regulate the intestinal microecology by increasing the abundance of potentially beneficial bacteria and decreasing the abundance of potentially harmful bacteria, thereby promoting gastrointestinal peristalsis.

### 2.4. ATTF Partially Restored the Loperamide-Induced Gut Metabolite Changes

The metabolites produced by the gut microbiota can modulate intestinal function through intestinal functional factors or intestinal neuroendocrine factors, thus participating in the development of constipation. A nontargeted metabolomic analysis of cecal contents was performed in this study. The shared gut metabolites between the three groups were 1801 and 1187 in positive and negative ion modes, respectively (Figure 4A,B). To investigate whether LOP and HAF had an effect on metabolites, principal component analysis (PCA) was used to observe that the cecum contents of CON, LOP and HAF were clearly separated (Figure 4C), macroscopically exhibiting differences in metabolites between groups. We identified 353 differentially expressed metabolites between CON and LOP, including 260 downregulated metabolites and 93 upregulated metabolites relative to the CON group in the LOP group (Figure 4D). A total of 792 differentially expressed metabolites were detected in LOP and HAF, including 189 downregulated metabolites and 603 upregulated metabolites relative to the LOP group in the HAF group (Figure 4E). The results showed that there was a significant difference in metabolite changes between the two groups.

To investigate the biological functions of the differentially expressed metabolites in depth, a Kyoto Encyclopedia of Genes and Genomes (KEGG) pathway enrichment analysis was performed. Three groups of differentially expressed metabolites were significantly enriched in a number of metabolic pathways, and the ten most significant pathways were screened, with the most significant pathways (*p* < 0.001) including flavonoid biosynthesis, biosynthesis of phenylpropanoids, arachidonic acid metabolism, isoflavonoid biosynthesis, and serotonergic synapse (Figure 4F). We focused on the enrichment of LOP and HAF differential metabolites, and the most significant metabolic pathways in Appendix A were also the six pathways described above. To verify this hypothesis, a KEGG topology analysis (Appendix A) was carried out, and the most significant metabolic pathways were also dominated by those metabolic pathways.

Among the abovementioned six pathways, the alteration of four metabolic pathways related to flavonoid synthesis might be directly affected by the addition of ATTF, while the enrichment of serotonergic synapse and arachidonic acid metabolism is closely related to the alleviation of constipation. Differential metabolites coenriched in serotonergic synapses and arachidonic acid metabolism included thromboxane B2 (TXB2), prostaglandin J2 (PGJ2), and prostaglandin-c2 (Figure 4G). 15-deoxy-d-12,14-PGJ2, 20-hydroxy-leukotriene B4, 6-ketoprostaglandin E1, and delta-12-prostaglandin J2 are differential metabolites enriched only in arachidonic acid metabolism (Figure 4H). Notably, butyric acid is an important short-chain fatty acid for relieving constipation (Figure 4I). The abundance of all the abovementioned nine metabolites decreased in the model group and increased in the HAF group. In conclusion, HAF significantly changed the important metabolic pathways of loperamide-induced STC mice, and ATTF treatment directly affected the serotonergic synapse pathway, which is closely related to the relief of constipation.

### 2.5. ATTF Enhanced the Expression of Serotonergic Synapse-Related Factors in Loperamide-Induced STC Mice

Serotonin (5-HT) is a key neurotransmitter involved in the regulation of intestinal motility and can speed up defecation. HAF restored the loperamide-induced decrease in serum 5-HT to high levels (Figure 5A). Inducible nitric oxide synthase (iNOS) affected smooth muscle relaxation by regulating the production of NO. We found that HAF significantly reduced the serum level of iNOS in STC mice (Figure 5B). In the serotonergic synapse metabolic pathway, 5-HT can bind to 5-HT receptor 2A (*5-HT_2A_*) and then activate Phospholipase A2 (*PLA_2_*) and Cyclooxygenase-2 (*COX2*), further stimulating the production of PGJ2 and TXB2. Transient receptor potential A1 (*TRPA1*) senses various stimulus signals in the intestinal lumen and mediates the release of 5-HT from intestinal enterochromaffin cells (ECCs), which ultimately activates Myosin light chain 3(*MLC3*), a key factor in smooth muscle cells, and promotes smooth muscle contraction. In the present study, the mRNA expression of *5-HT_2A_*, *PLA_2_*/*COX2*, *TRPA1*, and *MLC3* was significantly enhanced by HAF treatment in loperamide-induced STC mice (Figure 5D–G). In general, HAF could promote gastrointestinal motility by regulating serotonergic synapse-related factors.

### 2.6. Correlations between Specific Gut Microbes, Gut Metabolites, and Core Host Parameters

To elucidate the linkages between the factors involved in the improvement of constipation by HAF, we analyzed the correlations between host parameters, different gut microbiota, and metabolites. Correlations between the differential microbe taxa and host parameters are shown in Figure 6A. All 19 taxa had significant correlations with defecation indicators and serotonergic synapse-related factors. FN and FW, as important defecation indicators, showed strong correlations with all 19 taxa but only significantly positive correlations with *Lactobacillus* and *Bacillus* and strong negative correlations with the other 17 taxa. *COX2* and *PLA2* are important factors of the serotonergic synapse pathway. They showed the same correlation pattern with FN and FW. Again, they were only strongly positively correlated with *Lactobacillus* and *Bacillus* (Figure 6A). It is worth mentioning that Lachnospiraceae also showed significant correlations with nearly all the host indicators (Figure 6A). Interestingly, FBST only showed a significant positive correlation with Eubacterium_coprostanoligenes_group and *Anaerotruncus*. Obviously, the significant changes in high abundance taxa, such as *Lactobacillus*, *Bacillus,* and Lachnospiraceae, might contribute to ATTF promoting defecation in STC mice by modulating the serotonergic synapse pathway.

Correlations between differential intestinal metabolites and differential host parameters are shown in Figure 6B and Appendix A. We found that FN and FW were positively correlated with metabolites enriched in the serotonergic synapse pathway, including PGJ2, TXB2, 14,15-DiHETrE and PGC2. Both 5-HT and iNOS were strongly correlated with the differential metabolites, but the correlation trends were completely opposite. This suggests that differential metabolites involved in serotonergic synapses might play an important role in the amelioration of constipation, especially the interaction between TXB2, PGJ2, and 5-HT.

We also analyzed the correlation between differential microbial taxa and important differential metabolites (Figure 6C and Appendix A). Erysipelatoclostridiaceae, Oscillospiraceae, Lachnospiraceae, *Erysipelatoclostridium*, *Helicobacter*, *Anaerotruncus*, and *Lachnospiraceae_NK4A136_group* were negatively correlated with all five differential metabolites involved in serotonergic synapses, and only PGJ2 was significantly correlated with all nine taxa (Figure 6C). Notably, *Bacillus* and *Lactobacillus* were positively correlated with all five differential metabolites and had a significant correlation with PGJ2. The close relationship between these differential metabolites and microbes implies that they may play an important synergistic role in the ATTF laxative process.

## 3. Discussion

Loperamide is a highly effective drug used for acute diarrhoea. It acts on the gastrointestinal tract wall, inhibits the contraction of intestinal smooth muscle, reduces intestinal peristalsis, and reduces the release of neurotransmitters from intestinal nerve endings, directly inhibiting the intestinal peristalsis reflex. In this study, loperamide successfully produced a mouse model of STC. Previous studies revealed that ATAE was effective in relieving symptoms of loperamide-induced STC in mice [10]. We further found that flavonoid-rich ATES also successfully reversed the symptoms of STC, while ATEP did not. To this end, we purified flavonoids from ATES, validated the laxative efficacy of ATTF in a mouse model of STC, and explored the laxative mechanism of ATTF from the perspective of gut microbiota and intestinal metabolism.

The results showed that ATTF can maintain the intestinal microecology by increasing the abundance of beneficial bacteria such as *Lactobacillus* and *Bacillus* and reducing the dominant commensals, such as Lachnospiraceae, thus changing the intestinal microbial structure and composition; ATTF also altered the abundances of some intestinal metabolites enriched in the serotonergic synapse pathways. Importantly, ATTF increased the serum 5-HT content and the colonic mRNA expressions of *5-HT_2A_*, *PLA_2_* and *COX2*, which are involved in the serotonergic synaptic pathway. Interestingly, ATTF elevated the mRNA expression of TRPA1 (a stimulator of 5-HT release) and MLC3 (a stimulator of smooth muscle motility) and decreased the mRNA expression of iNOS (a synthase of the inhibitory neurotransmitter NO), thereby increasing the contraction of the colon and improving bowel motility, ultimately leading to defecation.

Flavonoids have abundant pharmacological activities and biological effects, such as antioxidation, prevention of cardiovascular disease and cancer, and reduction in inflammation [11], and flavonoids also have therapeutic effects on gastrointestinal disorders. Flavonol naringenin [12], 7,8-dihydroxyflavone [13], and *Allium mongolicum* Regel and its flavonoids [14] could reverse the symptoms of loperamide-induced constipation in mice. In addition, fermented soymilk with more total isoflavones had better laxative effects than unfermented soymilk [15], whereas isoflavone-deficient foods could cause constipation while reducing beneficial bacteria and regulating gut microbiota [16]. For the first time, we systematically studied the laxative effect of AT, AT extracts and ATTF, and all these previous studies provide additional support.

Several components with high abundance in ATTF were (+)-epicatechin, (−)-catechin, L-epicatechin, isoquercitrin, procyanidin B2, and (−)-epiafzelechin. Studies have shown that proanthocyanidins (PACs) are compounds formed by the polymerization of flavan-3-ol by C-C bonding as a structural unit, which can be divided into monomers, oligomers and polymers according to their polymerization degree; (+)-epicatechin, (−)-catechin, L-epicatechin, and (−)-epiafzelechin in ATTF are haploid; procyanidin B2 in ATTF is a dimer of oligomers; and B-type PACs are more common in most plants in nature.

Many foodborne PACs can be ingested by people in daily life and have strong biological activities. For example, PACs in peanut seed coats can lower blood sugar [17], PACs in apples have anticancer activity [18], and PACs in red wine can prevent cardiovascular disease [19]. It has been reported that grape seed PAC feeding can reduce the oxidative stress response of rat intestinal epithelial cells and relieve intestinal structural damage [20]. Grape seed PACs significantly increase the abundance of beneficial bacteria in bovine rumen [21]. Epicatechin, (−)-catechin, and isoquercitrin also have anti-inflammatory effects [22,23]. Therefore, we speculate that the laxative effect of ATTF could be related to the anti-inflammatory and intestinal barrier restoration effects of several monomeric components. In addition, quercetin, an analogue of isoquercitrin, also has the potential to prevent constipation induced by loperamide in rats [24]. Quercetin can promote gastrointestinal motility and mucin secretion in SD rats with loperamide-induced constipation by regulating mAChR downstream signaling [25].

Flavonoids are known to have beneficial effects in modulating the human gut microbiota, which in turn has a positive effect on health and has thus become a popular research concern [11]. We found that ATTF could effectively modulate the cecal microbiota of loperamide-induced STC mice. As the dominant genera, *Lactobacillus* and *Bacillus* increased in abundance with ATTF intervention and had an extremely strong positive correlation with the important phenotypic indicators FN and FW, demonstrating a close association with constipation and possibly a significant contribution to the relief of constipation.

Previous studies have shown that total plant flavonoids and several different types of flavonoids, such as flavonols, flavanols, anthocyanins and flavones, can increase the abundance of *Lactobacillus* spp. [11]. For example, soy isoflavones promote the intestinal microorganism *L. mucosae* [26]. Anthocyanins can increase the abundances of *Bifidobacterium* spp., *Lactobacillus* spp. and *Akkermansia* in the intestinal tract [27]. PACs, the main component of ATTF, also acts as a prebiotic, promoting the growth of gut microbiota such as *Lactobacillus*, *Bifidobacteria*, and butyrate-producing bacteria, and the two-way regulation between the two not only improves the bioavailability of PACs but also helps the growth of intestinal beneficial bacteria [28].

The combination of lotus PACs and *L. casei*-01 enhances the physiological activity of lotus PACs by increasing the level of total antioxidant capacity, thereby improving learning and memory impairment, and intestinal function has also been reported to be regulated by the above two-way action [28]. The interaction between common *Lactobacillus* and lychee peel PACs has a two-way regulatory effect. PACs can effectively regulate the growth of *Lactobacillus*, and conversely, *Lactobacillus* also promotes the metabolism of PACs [29,30]. Grape seed PACs can reduce gastrointestinal mucosal damage caused by oxidative stress and improve the mucosal barrier through antihistamine action and regulation of intestinal microflora [31,32], and the combination of PACs and probiotic preparations can improve intestinal damage in mice with diarrhoea [33].

In recent years, reports of constipation relief by beneficial bacteria such as *Lactobacillus* and *Bacillus* or their microbial preparations have also become increasingly topical. *L. rhamnosus* strains alleviate loperamide-induced constipation through different pathways independent of short-chain fatty acids [34], and lyophilized powder of *L. paracasei* alleviates loperamide-induced constipation and improves gastrointestinal function by increasing SCFAs and interstitial cells of Cajal (ICCs) [35]. *Bacillus coagulans* was effective in alleviating the adverse effects of loperamide-induced constipation in Kunming mice [36]. *B. subtilis* alleviated the constipation induced by difenoxylate in STC mice [37].

Flavonoids are mainly metabolized in the gastrointestinal tract. During metabolism, flavonoids can be broken down into other metabolites, which are absorbed in the stomach and intestine and then play a local role in epithelial cells, endocrine cells, immune cells and the intestinal lumen of the gastrointestinal tract. A significant portion of the flavonoids reach the colon, where they interact with the gut flora. Microbes in the colon can convert flavonoids into small molecules that can enter the circulatory system and exert systemic effects [38].

We found that ATTF significantly altered the metabolite composition of mouse cecal contents, and a large part of the differential metabolites were enriched in the flavonoid biosynthesis, biosynthesis of phenylpropanoids, and isoflavonoid biosynthesis pathways. Among differential metabolites, some components have a role in repairing the intestinal barrier. For example, the (−)-epicatechin can repair inflammatory bowel disease and high-fat diet-induced intestinal barrier damage [39,40], and apigenin and pine genistein can effectively repair the intestinal barrier and significantly improve the damaged colon tissue and thus treat ulcerative colitis in mice [41,42]. Naringenin, nobiletin, and hesperetin showed modulating effects on ulcerative colitis by protecting the integrity of the colonic mucosal layer [43]. Isoliquiritigenin repairs pancreatic injury and intestinal dysfunction after severe acute pancreatitis through the Nrf2 signaling pathway [44]. We speculate that ATTF might also play a laxative role by improving the gut barrier, although we did not confirm this.

In addition, we identified a series of differential metabolites in the arachidonic acid metabolic pathway through metabolomics technology, including PGJ2, delta-12-prostaglandin J2, 15-deoxy-d-12, 14-PGJ2 and 6-ketoprostaglandin E1, TXB2, 11-dehydro-thromboxane B2 and 20-hydroxy-leukotriene B4. ATTF intervention increased the abundance of each of these metabolites. Studies have shown that 15-deoxy-d-12, 14-PGJ2 can effectively contract uterine smooth muscle [45,46]. Synthetic analogues of prostaglandin E1 are effective in the treatment of methotrexate-induced intestinal mucosal injury in rats [47]. TXB2, an inactive metabolite of thromboxane A2 (TXA2), is often used as an indicator to detect TXA2 production, which is at reduced levels in female patients with STC [48]. TXA2 also has an activating regulatory effect on pacemaker Cajal interstitial cell activity [49]. It is possible that leukotrienes (LTs) play an important role in the regulation of colonic motility following pathological stimulation, and they have been shown to contract the smooth muscle of the large intestine in vitro [50].

Meanwhile, we found that some differential metabolites were enriched in both arachidonic acid metabolism and serotonergic synapses, such as PGJ2 and TXB2. PGJ2 and TXB2 may have a contracting effect on colonic smooth muscle. 5-HT is the core factor of serotonergic synapses and plays an important role in regulating intestinal motility and intestinal secretion [51]. In addition, 5-HT can activate the peristaltic reflex by activating 5-HT receptors at the mucosal end of intrinsic primary afferent neurons [52,53]. Previous studies have shown that *B. subtilis* could promote the release of 5-HT via bile acid and its receptor TGR5 pathway to regulate gastrointestinal motility in STC mice [37]. *Bifidobacterium* and *Lactobacillus* can promote 5-HT secretion from ECCs in constipated mice [54]. We found that the expression of serum 5-HT and *5-HT_2A_* was downregulated with the development of constipation and recovered with the intervention of ATTF, and serum 5-HT showed a positive correlation with *Lactobacillus* and *Bacillus.*

5-HT is mostly activated and secreted by enterochromaffin cells (ECCs) of the intestinal mucosal epithelium under chemical or mechanical stimulation and further activates gastrointestinal peristalsis [55]. As an important sensor receptor molecule for the release of 5-HT from ECCs, TRPA1 can indirectly regulate gastrointestinal function via 5-HT [56]. In mouse colonic EC cells, IL-33 signaling can selectively activate TRPA1 channels in ECCs via PLC-γ1 to promote the inward flow of Ca^2+^ and 5-HT release [57]. In the small intestine, chemicals can activate TRPA1 to induce 5-HT release from ECCs to modulate gastrointestinal motility; exogenous agonists from the lumen or endogenous agonists from mucosal tissues can activate TRPA1 and induce PGE2 release from ECCs, thereby modulating colorectal motility [58]. Previous studies have shown that TRPA1 can be activated by flavonoids. Isoflavone and chalcone compounds displayed potent TRPA1 agonistic activity [59]. Gingerol derived from ginger might improve digestive function through serotonin release from ECCs by inducing TRPA1-mediated calcium influx [60]. In this study, *TRPA1* was significantly inhibited by loperamide, but ATTF effectively restored and significantly elevated the expression of *TRPA1*. This finding suggests that ATTF might promote the release of 5-HT from ECs by stimulating TRPA1.

In addition, *5-HT_2A_*, *PLA_2_* and *COX2* are located between 5-HT and PGJ2/TXB2 in the serotonergic synapse pathway. We found that ATTF intervention could significantly upregulate their mRNA expression, thereby increasing the abundances of PGJ2 and TXB2. Importantly, we also found that ATTF intervention significantly increased the expression of MLC3, a biomarker of smooth muscle contraction.

NO can induce smooth muscle relaxation and impair gastrointestinal motility [61]. Inducible nitric oxide synthase (iNOS) is a key enzyme in the synthesis of the inhibitory neurotransmitter nitric oxide (NO). Previous studies have shown that the main component of ATTF, (−)-epicatechin, prevents LPS-induced renal inflammation and reduces the expression of iNOS [62]; PGJ2 blocks iNOS expression and subsequent NO production via a PPARγ-mediated mechanism in thylakoid cells [63]. These results support our findings regarding iNOS.

In this study, we demonstrated the alleviation effect of AT extracts on STC by using male mice but did not explore it in female mice, so it is worthy of being further explored. In addition, the results of this study confirm the important role of gut microbiota in the laxative effect of ATTF, but whether gut microbiota and its metabolites mediate the laxative effect of ATTF still needs to be demonstrated by fecal microbiota transplantation and antibiotic experiments. In summary, we systematically explored the active components of AT in alleviating symptoms through loperamide-induced STC mice and its effects on host parameters, gut microbiota, and gut metabolites. We found that ATAE can significantly alleviate the symptoms of STC caused by loperamide, where the active fraction that plays a key role is ATES, not ATEP. Flavonoids are the main active components of ATES. Purified flavonoids from ATES (ATTF) also exert a good laxative ability. ATTF might relieve constipation by regulating the gut microbiota (e.g., *Lactobacillus* and *Bacillus*) and the expression of differential metabolites enriched in serotonergic synaptic metabolic pathways (e.g., PGJ2 and TXB2), as well as by increasing serotonergic synaptic pathway-related factors (e.g., 5-HT) to restore intestinal homeostasis and promote gastrointestinal motility. The complex relationships among gut microbiota, gut metabolites, and defecation phenotype indicators have helped us to understand the laxative mechanism of ATTF. These results not only lay the foundation for the application of AT and its active components in purgatives but also provide a reference for exploring the laxative mechanism of plant-derived natural products.

## 4. Materials and Methods

### 4.1. Preparation of AT Extract

ATAE: AT fruits were obtained from Gongshan Yongjia Agricultural Development Co., Ltd., in Nujiang City, Yunnan Province (China). AT powder is crushed with a pulverizer, AT powder (1000 g) was boiled for 3 min in ultrapure water (the material/liquid ratio was 1:9). The extraction solution was immediately centrifuged at 5000× *g* rpm for 5 min after cooling to ambient temperature. The precipitates were collected twice under the same circumstances and then thrown away. The supernatants were all mixed and vacuum freeze-dried for two to three days. The yield of the AT water extract (ATAE, 133 g) was 13.3%.

ATES and ATEP: 120 g ATAE was added to 70% ethanol (solid/liquid ratio = 1:24), left to stand for 4 h, and centrifuged at 5000× *g* rpm for 5 min. The supernatant and precipitate were collected and freeze-dried for 2–3 days, and the final yields of the AT ethanol-soluble part (ATES, 55.2 g) and AT ethanol precipitation (ATEP, 51.6 g) were 46% and 44%, respectively.

ATTF: First, the resin was activated, and then static adsorption was performed. The column material was added with a pH value of 4.0 and a concentration of 2 mg/mL (50 g ATES and 25 L ultrapure water) ATES (material/liquid ratio of 1:15) and soaked for 6 h (shaker 25 °C, 120 rpm). The column material was eluted, the adsorbed column material was poured into the column, and then 7 BV of 40% ethanol was added for elution. Finally, the ethanol was removed using a rotary evaporator and freeze-dried for 2–3 days, giving a yield of 20% for AT total flavonoids (ATTF, 10 g).

### 4.2. Animal Experimental Design

Six-week-old male Kunming mice (20–25 g) were purchased from Chengdu Dossy Experimental Animals Co., Ltd., Chengdu, China. All mice were kept under particular pathogen-free barrier settings (24 ± 1 °C, 30–50% humidity, 12 h of light, lights out at 20:00), and they were fed a standard chow diet with water available at all times and 10.8% fat, 68.7% carbs, and 20.5% protein. To establish the STC mouse model, 6 mg/kg·bw Loperamide (loperamide, mg/kg·body weight, Sigma, St. Louis, MI, USA) was employed. Loperamide was given to STC model mice for 8 days to induce STC, and the mice received daily oral dosages of 300 μL of solutions that were assigned to the appropriate experimental group. The Maren pill (Beijing Tongrentang Pharmaceutical Co., Ltd., Beijing, China), ATAE, ATES, and ATEP dosages were dissolved in 300 μL of saline solution (Figure 1A,C,E and Figure 2A). A total of four animal experiments were carried out in this study, and the group settings are described as follows.

Animal experiment 1 (Figure 1A): The mice were separated into 6 groups of 12 mice each after 7 days of acclimatization. CON group (control group, received saline solution as vehicle), LOP group (model group, received 6 mg/kg·bw loperamide), POS group (positive control group, received 6 mg/kg·bw loperamide and 900 mg/kg·bw Maren pill), LAE group (received 6 mg/kg·bw loperamide and 500 mg/kg·bw ATAE, low dosage), MAE group (received 6 mg/kg·bw loperamide and 750 mg/kg·bw ATAE, medium dosage), and HAE group (received 6 mg/kg·bw loperamide and 900 mg/kg·bw ATAE, high dosage).

Animal experiment 2 (Figure 1C): The mice were separated into 6 groups of 12 mice each after 7 days of acclimatization. CON group, LOP group, MAE group, LES group (received 6 mg/kg·bw loperamide and 250 mg/kg·bw ATES, low dosage), MES group (received 6 mg/kg·bw loperamide and 375 mg/kg·bw ATES, medium dosage), and HES group (received 6 mg/kg·bw loperamide and 500 mg/kg·bw ATES, high dosage).

Animal experiment 3 (Figure 1E): The mice were separated into 6 groups of 12 mice each after 7 days of acclimatization. CON group, LOP group, MAE group, LEP group (received 6 mg/kg·bw loperamide and 250 mg/kg·bw ATEP, low dosage), MEP group (received 6 mg/kg·bw loperamide and 375 mg/kg·bw ATEP, medium dosage), and HEP group (received 6 mg/kg·bw loperamide and 500 mg/kg·bw ATEP, high dosage).

Animal experiment 4 (Figure 2A): The mice were separated into 4 groups of 12 mice each after 7 days of acclimatization. CON group, LOP group, LAF group (received 6 mg/kg·bw loperamide and 150 mg/kg·bw ATTF, low dosage), and HAF group (received 6 mg/kg·bw loperamide and 300 mg/kg·bw ATTF, high dosage).

All mice were deprived of food but not water overnight for 10 h before the defecation test and gastrointestinal transit test. All procedures were previously approved by the Animal Ethics Committee of Yunnan Agriculture University.

### 4.3. Defecation Test

On the seventh day of the animal experiment, mice received their regular dosage of administration via gavage at 08:00. Thirty minutes later, handmade ink (300 μL) was administered by gavage, and the mouse defecation experiment was formally launched. FBST was meticulously noted for each mouse, and for six hours following the commencement of the defecation test, FW, FN, and fecal water content were all examined to determine the laxative effect.

### 4.4. Gastrointestinal Transit Test and Tissue Collection

On the eighth day of the animal experiment, mice received the standard dose of the administration at 08:00. The ink (300 μL) was administered to each mouse thirty minutes later. The mice were swiftly euthanized in a room filled with CO_2_ after 20 min, and the abdominal cavity was opened to collect blood from the abdominal aorta. Each digestive tract’s mesentery was meticulously removed, and the lengths of the whole small intestine and the area designated by the ink were measured to determine GTR.

Each mouse’s proximal colon, distal ileum, and cecum were precisely dissected at the same time. To eliminate feces, the ileum and colon segments’ contents were extensively flushed with cold PBS. The cecum contents were removed and rinsed in a 2 mL Eppendorf tube with 1.0 mL of ice-cold Milli-Q water. The cecum, kidney, and liver tissues were weighed. Then, all tissues and cecal contents were flash frozen in liquid nitrogen and kept at −80 °C pending analysis.

### 4.5. Phytochemical Composition Determination by Widely Targeted Metabolomics

After the AT sample was pretreated [64], an EXIONLC System was used for UHPLC separation (Sciex). Formic acid at 1% in water served as mobile phase A, and acetonitrile served as mobile phase B. The temperature in the column was fixed at 40 °C. The injection volume was 2 L, and the autosampler’s temperature was set at 4 °C. For the development of the test, a Sciex QTrap 6500+ (Sciex Technologies) was used. The typical ion source settings were as follows: ion spray voltage: +5500/−4500 V; curtain gas: 35 psi; temperature: 400 °C; ion source gas 1:60 psi; ion source gas 2:60 psi; and DP: 100 V. Finally, SCIEX Analyst Work Station Software (Version 1.6.3) was employed for MRM data acquisition and processing. An in-house R program and database were applied for peak detection and annotation.

### 4.6. 16S rRNA Gene Sequencing and Analysis of Cecal Contents

Following the manufacturer’s instructions, microbial community genomic DNA was isolated from cecal content using the E.Z.N.A. soil DNA kit (Omega Biotek, Norcross, GA, USA). The DNA extract was examined on a 1% agarose gel, and a NanoDrop 2000 UV—vis spectrophotometer was used to measure the DNA concentration and purity (Thermo Scientific, Wilmington, DE, USA). An ABI GeneAmp^®^ 9700 PCR thermocycler was used to amplify the hypervariable region V3-V4 of the bacterial 16S rRNA gene using the primer pair 338F (5′-ACTCCTACGGGAGGCAGCAG-3′) and 806R (5′-GGACTACHVGGGTWTCTAAT-3′) (ABI, Carlsbad, CA, USA). The 16S rRNA gene was amplified using PCR. The reaction system and reaction procedure were based on a previous method [65]. The AxyPrep DNA Gel Extraction Kit (Axygen Biosciences, Union City, CA, USA) was used to extract the PCR product from a 2% agarose gel, purify it as directed by the manufacturer, and quantify it using a QuantusTM Fluorometer (Promega, Madison, WI, USA) for Illumina MiSeq sequencing. Purified amplicons were pooled in equimolar amounts and paired-end sequenced on an Illumina MiSeq PE300 platform/NovaSeq PE250 platform (Illumina, San Diego, CA, USA) according to the standard protocols by Majorbio Bio-Pharm Technology Co., Ltd., (Shanghai, China). Based on the ASVs information, rarefaction curves and alpha diversity index (Simpson index) were calculated with Mothur v1.30.1. The similarity among the microbial communities in different samples was determined by principal coordinate analysis (PCoA) based on binary Euclidean distance using Vegan v2.5-3 package. The PERMANOVA test was used to assess the percentage of variation explained by the treatment along with its statistical significance using Vegan v2.5-3 package. The linear discriminant analysis (LDA) effect size (LEfSe) [9] (http://huttenhower.sph.harvard.edu/LEfSe, accessed on 10 January 2023) was performed to identify the significantly abundant taxa (phylum to genera) of bacteria among the different groups (LDA score > 2, *p* < 0.05) [65]. The raw reads were deposited into the NCBI Sequence Read Archive (SRA) database (Accession Number: PRJNA942344).

### 4.7. Untargeted Metabolomics of Cecal Contents

Pretreated cecal contents were carefully transferred to sample vials for LC—MS/MS analysis. The instrument platform for this LC—MS analysis is the UHPLC-Q Exactive HF-X system manufactured by Thermo Fisher Scientific. Two microliters of the sample were separated by an HSS T3 column (100 mm × 2.1 mm i.d., 1.8 μm) and then subjected to mass spectrometry detection. During the period of analysis, all samples were stored at 4 °C. The mass spectrometric data were collected using a Thermo UHPLC-Q Exactive HF-X mass spectrometer equipped with an electrospray ionization (ESI) source operating in either positive or negative ion mode. Data acquisition was performed in data-dependent acquisition (DDA) mode. The detection was carried out over a mass range of 70–1050 *m/z*.

The response intensity of the sample mass spectrum peaks was normalized by the sum normalization method. At the same time, variables with relative standard deviation (RSD) > 30% of QC samples were removed, and log10 transformation was performed to obtain the final data matrix for subsequent analysis. The R package ropls (Version 1.6.2) was used to perform principal component analysis (PCA) and orthogonal least partial squares discriminant analysis (OPLS-DA). In addition, Student’s *t*-test and fold difference analysis were performed. The selection of significantly different metabolites was determined based on the variable importance in the projection (VIP) obtained by the OPLS-DA model and the *p* value of Student’s t-test, and the metabolites with VIP > 1 and *p* < 0.05 were significantly different metabolites.

### 4.8. Enzyme-Linked Immunosorbent Assay

After the mice were sacrificed, their blood was collected immediately, incubated at 37 °C for 30 min, centrifuged at 4 °C at 3500× *g* rpm for 10 min, and serum was collected. Serotonin (5-HT) and induced nitric oxide synthases (iNOS) were determined by using the ST/5-HT (Serotonin/5-Hydroxytryptamine) ELISA Kit Instruction (Sangon Biotech, Shanghai, China) and Mouse NOS2/iNOS (Nitric Oxide Synthase 2, Inducible) ELISA Kit Instruction (Sangon Biotech, Shanghai, China), respectively.

### 4.9. RNA Preparation and Quantitative PCR Analysis of Gene Expression

A TaKaRa MiniBEST Universal RNA Extraction Kit (9767, Takara, Japan) was used for total RNA extraction from the mouse colon. A PrimeScript™ RT reagent Kit with gDNA Eraser (Perfect Real Time, RR047A, Takara, Osaka, Japan) was used for RNA reverse transcription, and TB Green^®^ Premix Ex Taq^TM^ II (Tli RNaseH Plus, RR820A, Takara, Osaka, Japan) was used for the quantitative PCR analysis of gene expression. The relative amount of the target mRNA was normalized to the *RPL-19* level, and the results were calculated by the 2^−∆∆Ct^ method. The primer sequences are presented in Appendix A.

### 4.10. Statistical Analysis

The means ± standard errors of the means (SEMs) are used to express the data. To evaluate two independent groups, Student’s unpaired two-tailed *t*-test was used. Spearman’s *r* coefficients were used to construct bivariate correlations. *p* values were adjusted for multiple testing using R language and according to the *BH* procedures. HemI 1.0 software was used to create heatmaps (http://hemi.biocuckoo.org/down.php, accessed on 15 January 2023). Results with a *p* value < 0.05 were deemed statistically significant unless otherwise noted in the figure legends.

## Figures and Tables

**Figure 1 ijms-24-07191-f001:**
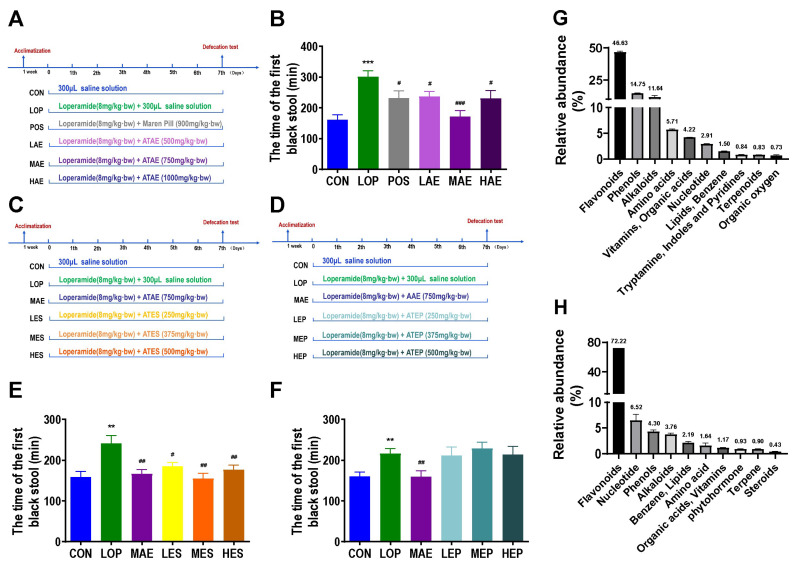
The laxative active fraction of ATAE and its main chemical composition. Grouping and basic workflow for animal experiments 1 (**A**), 2 (**C**) and 3 (**D**). The defecation time of the first black stool (FBST) in animal experiments 1 (**B**), 2 (**E**), and 3 (**F**). Chemical composition of ATES (**G**) and ATTF (**H**) based on widely targeted metabolomics. The data are expressed as the means ± SEMs (*n* = 10–12). **, ***, compared with the CON group; #, ##, ###, compared with the LOP group. **, *p* < 0.01; ***, *p* < 0.001. #, *p* < 0.05; ##, *p* < 0.01; ###, *p* < 0.001.

**Figure 2 ijms-24-07191-f002:**
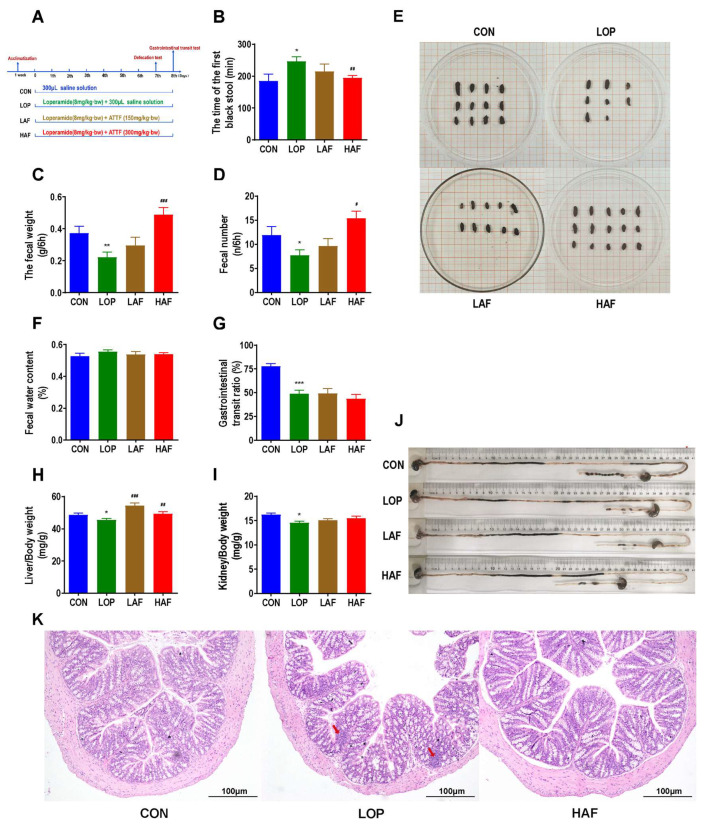
Influence of ATTF on loperamide-induced constipation symptoms in mice. (**A**) Grouping and basic workflow of animal experiment 4. (**B**) The defecation time of the first black stool, FBST. (**C**) Wet weight of feces excreted in 6 h, FW. (**D**) Number of feces excreted in 6 h, FN. (**E**) Representative fecal morphology of each group. (**F**) Fecal water content. (**G**) Gastrointestinal transit rate, GTR. (**H**) Liver index. (**I**) Kidney index. (**J**) Ink advancing distance and intestinal length. (**K**) Photomicrographs of H&E-stained proximal colon sections. Infiltration of inflammatory cells into the tissue is shown in blue in the photomicrographs in the LOP group. The data are expressed as the means ± SEMs (*n* = 10–12). *, **, ***, compared with the CON group; #, ##, ###, compared with the LOP group. *, *p* < 0.05; **, *p* < 0.01; ***, *p* < 0.001. #, *p* < 0.05; ##, *p* < 0.01; ###, *p* < 0.001.

**Figure 3 ijms-24-07191-f003:**
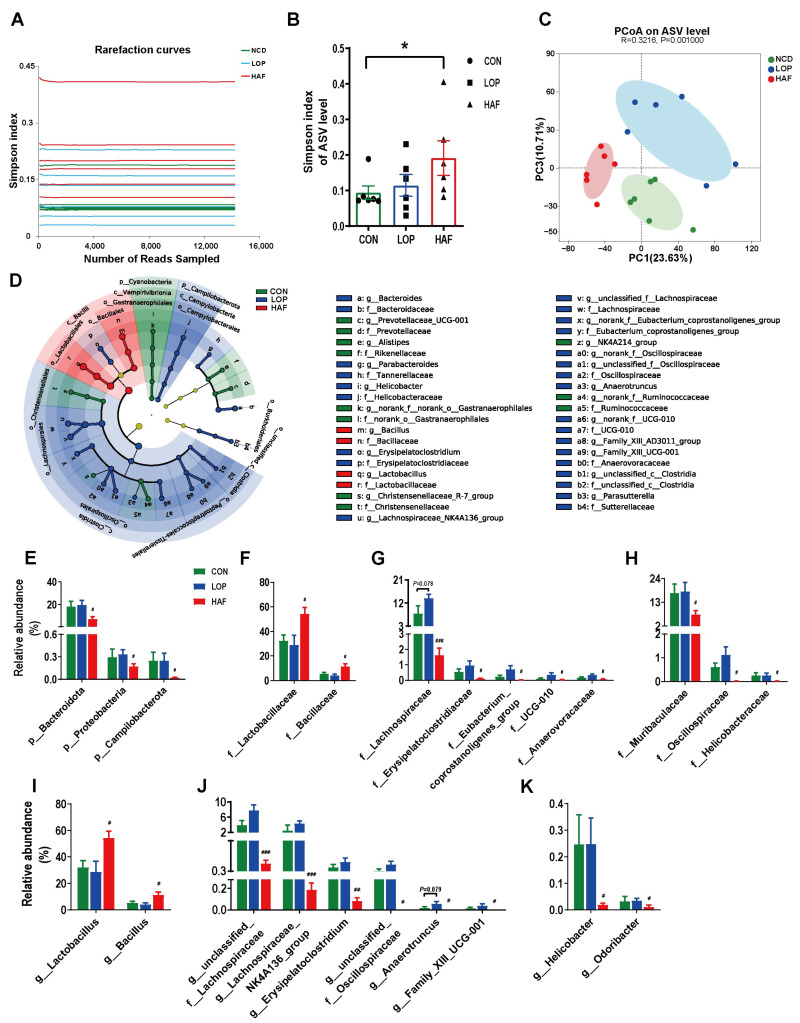
HAF restores the loperamide-induced gut microbial community structural and compositional shift. (**A**) The rarefaction curve of the Simpson index of each sample plateau at the ASV level. (**B**) Alpha diversity estimated by the Simpson index. (**C**) PCoA (principal coordinate analysis) plot based on binary Euclidean distance. (**D**) Linear discriminant analysis effect size (LEfSe) analyses (LDA score > 2.0). (**E**–**K**) Relative abundances of gut microbiota at the phylum, family, and genus levels, which were significantly affected by LOP or HAF, especially those reversed by HAF treatment. The data are expressed as the means ± SEMs (*n* = 6). *, compared with the CON group; #, ##, ###, compared with the LOP group. *, *p* < 0.05; #, *p* <0.05; ##, *p* < 0.01; ###, *p* < 0.001.

**Figure 4 ijms-24-07191-f004:**
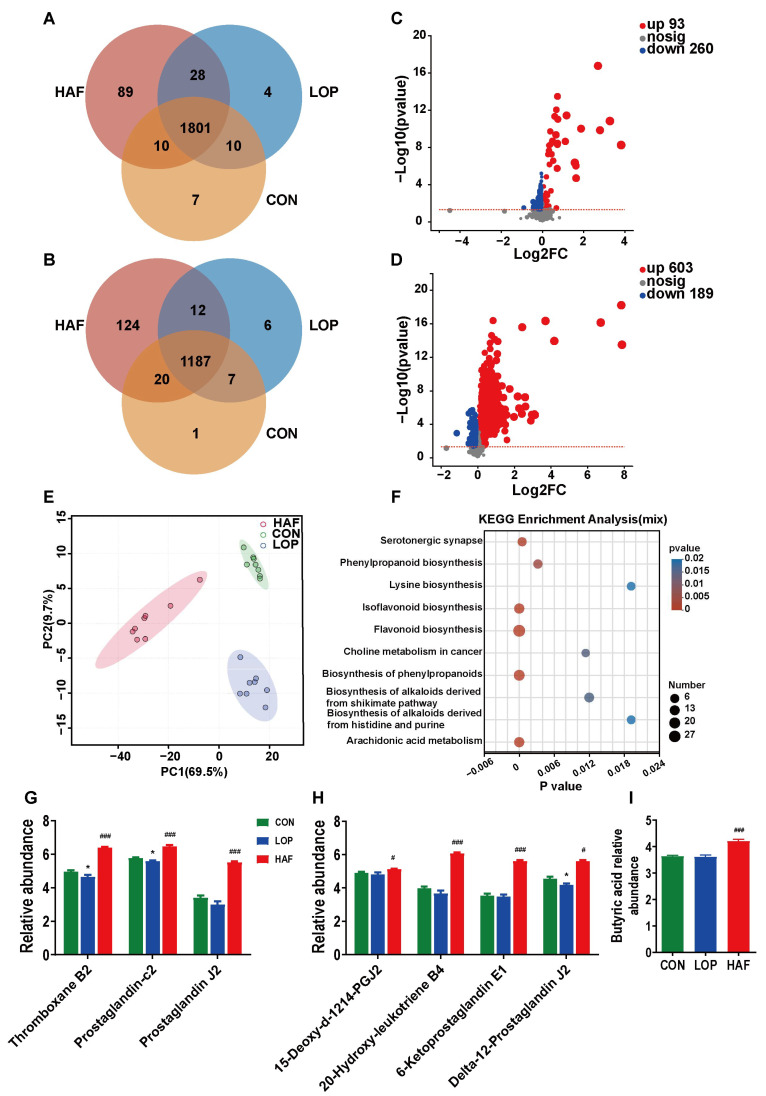
HAF changes the composition of loperamide-induced gut metabolites. Differential metabolites in the Venn diagram based on (**A**) cation detection and (**B**) anion detection. (**C**) Volcano map based on the LOP and CON groups. (**D**) Volcano map based on the LOP and HAF groups. (**E**) PCoA (principal coordinate analysis) based on all identified metabolites. (**F**) KEGG enrichment analysis. (**G**,**H**) Relative abundances of gut metabolites at the serotonergic synapse and arachidonic acid metabolism, respectively, which were significantly affected by loperamide or ATTF. (**I**) Relative abundances of butyric acid. The data are expressed as the means ± SEMs (*n* = 6). *, compared with the CON group; #, ###, compared with the LOP group. *, *p* < 0.05; #, *p* < 0.05; ###, *p* < 0.001. Log2FC, multiple change value of the expression difference of metabolites between the two groups; −Log10(*p* value), statistical test value of the difference in the expression of metabolites. The higher the value, the more significant the expression difference. *p* < 0.05, *n* = 8 in each group.

**Figure 5 ijms-24-07191-f005:**
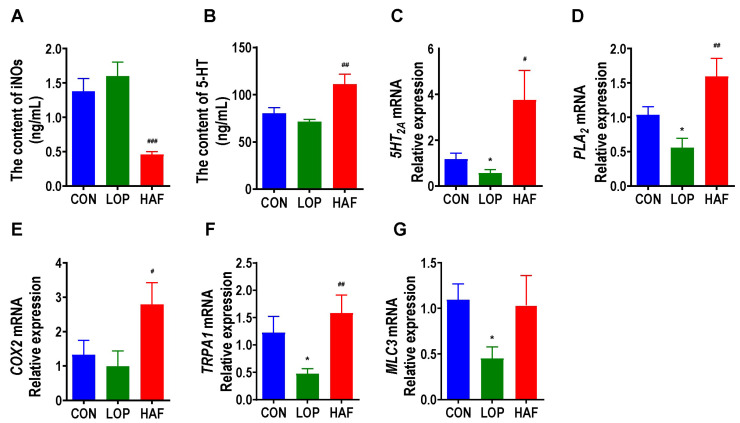
HAF increased the expression of gastrointestinal motility-related factors in the colon of STC mice. (**A**,**B**) Induced nitric oxide synthases (iNOS) and serotonin (5-HT) in serum. (**C**–**G**) The mRNA expression of serotonin 2 receptor (*5-HT_2A_*), phospholipase A2 (*PLA_2_*), cyclooxygenase-2 (*COX2*), transient receptor potential cation channel, subfamily A, member 1 (*TRPA1*), and Myosin light chain 3(*MLC3*) in the colon. The data are expressed as the means ± SEMs (*n* = 8). *, compared with the CON group; #, ##, ###, compared with the LOP group. *, *p* < 0.05. #, *p* < 0.05; ##, *p* < 0.01; ###, *p* < 0.001.

**Figure 6 ijms-24-07191-f006:**
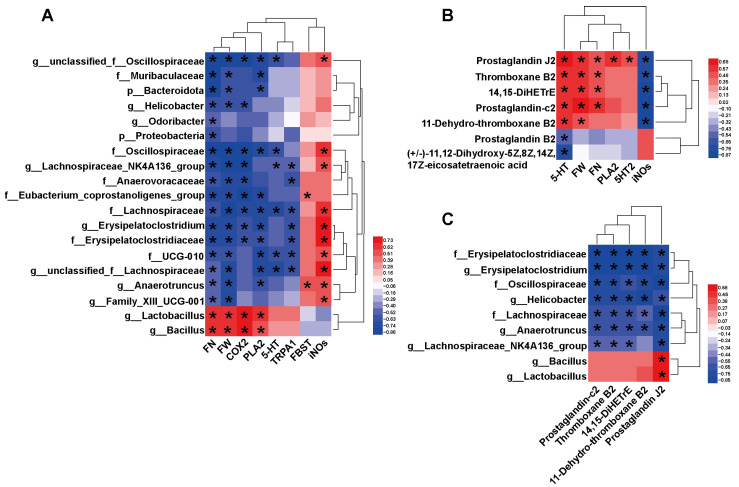
Heatmaps showing bivariate correlations between specific gut microbiota, gut metabolites and core host parameters in STC mice. (**A**) Correlations between the differential microbe taxa and differential host parameters. (**B**) Correlations between differential intestinal metabolites in serotonergic synapses and differential host parameters. (**C**) Correlations between the differential microbe taxa and differential intestinal metabolites in serotonergic synapses. The color at each intersection indicates the value of the *r* coefficient; *p* values were adjusted for multiple testing according to the *BH* procedures. * indicates a significant correlation between these two parameters (*p* < 0.05, *n* = 6 in each group).

## Data Availability

The raw reads of 16S rRNA gene sequence data were deposited into the NCBI Sequence Read Archive (SRA) database under BioProject accession number PRJNA840843. The raw reads of untargeted metabolomics data were deposited into the Mendeley Data database under BioProject accession number 10.17632/nbw9r97yph.1.

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
