# Peer review of "Flavonoids in Amomum tsaoko Crevost et Lemarie Ameliorate Loperamide-Induced Constipation in Mice by Regulating Gut Microbiota and Related Metabolites"

_ijms, 2023, doi:10.3390/ijms24087191_

Round 1
Reviewer 1 Report
I have read the draft entitled ‘‘Flavonoids in Amomum tsaoko Crevost et Lemarie ameliorate loperamide-induced constipation in mice by regulating gut microbiota and related metabolites’’ and I believe that it is a potentially publishable contribution. I have the following comments for improvement of the draft.
1. Introduction
L61-65: Please provide some citations to support these statements.
L71: You need to cite studies which indicate adverse effects of these drugs, and possibly the potential driving factors for departure from these conventional medicines (e.g., in remote communities access to these/the cost of the medicines may be unaffordable/cultural acceptability of herbal medicine).
L90-94: ‘‘We found that the ethanol-soluble part of ATAE is the key active site and is rich in flavonoids. Flavonoids have rich biological activities, including antioxidant, antibacterial, antiseptic, antidiabetes and anti-ovarian cancer effects [11], and thus have great potential to promote defecation’’. These are results or conclusions/ discussion, and should by implication, be moved to somewhere else other than the introduction.
2. Results and Discussion
L98: of ATAE laxation >> involved in ATAE laxative activity
L107-114: Most abbreviations used in this part were not expanded at first use in the manuscript.
3. Materials and Methods
L506: Revise 2.1. to 4.1.
L507-525: For the work to be repeatable by other researchers, it is important to indicate the exact quantities measured, in addition to the ratios presented.
4. General comment
The font style used in the manuscript should be harmonized.
Reviewer 2 Report
Dear Authors,
Congratulation and well done on your extensive work. I have read your paper with high interest, and I applaud your attention to detail.
I only had a few grammatical/ writing errors to report, and am missing a citation or two right at the start of the introduction, but else I have nothing to report.

Reviewer 3 Report
Flavonoids in Amomum tsaoko Crevost et Lemarie ameliorate loperamide-induced constipation in mice by regulating gut microbiota and related metabolites
This review aims to discuss the effect of amomum tsaoko crevost et lemarie on gut microbiota and constipation in a loperamide-induced constipated murine mouse model. Overall, it's a well-designed experiment and well-presented. Suggested revisions will improve the overall status of this manuscript.
Introduction
Line 89: In the introduction, you mentioned that the ethanol-soluble part of ATAE is the key active site and is rich in flavonoids. Is it from your preliminary data or published data? If so, include the citation.
What is your hypothesis?
LOP has not been defined in the introduction.
Results:
163: ATTF partially restored loperamide-induced gut microbial dysbiosis. What do you mean by "partially"?
172: In the α-diversity analysis, LOP and HAF were slightly lower than CON (Figure3B); CON and LOP were clearly separated in the PCoA plot of β-diversity analysis, with HAF between them (Figure 3C). Are these significant? If so, using what test?
Fig B: A box plot with data points would be more effective.
294: How did you perform the correlation?
Discussion:"
Some results have been repeatedly mentioned in the discussion section and vice versa.
What are the strengths, limitations and future directions of this study?
Need a summary paragraph in the discussion.
Methods:
508: How was the AT powder produced? How do you know that heating or any other preparatory method did not destroy the ethanol-soluble part of ATAE?
Line 536: What is the Maren tablet ect.? If they were not explained before, it has to be defined.
540-44: In the animal experiment 1, its not very clear about 6 groups? What were the dosages of LOP e.g. in LOP vs LAE group? Would that be more effective if you mentioned low, medium and high dose in the experimental groups to increase the clarity?
What is the difference between exp 2 and 3? It needs to be clearer. Consider Mentioning changing parameters in each experiment to avoid confusion.
556: All mice were deprived of food for how long?
569: what medicines?
610: SRA: PRJNA942344 was not found.
What analysis did you perform to analyse sequencing data? Were the functions predicted? Specific details of the bioinformatic analysis should be included under methods.
Consider having a more specific conclusion.
Reviewer 4 Report
Hu et al. have analyzed the active component of Amomum tsaoko that alleviates constipation in a mice model and its effects on microbiota and metabolites. The work is sound and I have a few questions to make it easier to follow the manuscript and complete it.
1) Since the material and methods are at the end, the start of results is rough. Please, explain that experiments 1, 2 and 3 (lines 99-100) are different set-ups and so on. In addition, please, make sure that the first time that a code is used is explained. e.g. CON, FBST or LOP (lines 109-110); and in some parts, it could be better to use “loperamide” rather than LOP (e. g. discussion). Please, check the use of code or full meaning accordingly. The provided abbreviation table is useful, but going forward and back to follow the text is not the best for the reader.
2) The role of various bacteria, metabolites and genes is profusely discussed, but it is not proposed a clear directionality, sometimes suggested but not clearly e. g. lines 349-353). Do you think that ATTF directly changes the microbiota, the microbiota the metabolites and the metabolites the host gene expression? Or ATTF changes host gene expression that changes the microbiota and that changes the metabolites? Or that the effect of ATTF is independent for each group?
3) In addition, some results obtained in this work are different from what is discussed. For example, in lines 468-469 is stated that “Previous studies have shown that B. subtilis could promote the release of 5-HTvia bile acid and its receptor TGR5 pathway to regulate gastrointestinal motility in STC mice [54]”, but in the results of correlation, there is no relationship between B. subtilis and 5-HT. Discrepancies like this and others should be properly discussed.
4) The discussion is lacking a clear statement of the limitations of the work (e.g. there has been used only one gender of mice, and the results can be generalized?) and the next steps that should be done to validate those results.
5) In material and methods, it is explained that 6 groups of 12 mice have been used. In each group of experiments, were they new mice? For example, CON group of experiments 1 and 2 are different? It should be clarified.
6) In material and methods, in the section about microbiota (lined 592-610), it is not explained how the analysis of diversity and taxonomic assignment have been done.
7) In material and methods, when correlations are explained (line 650) it is not stated that correction for multiple tests has been used, as can be concluded from Figure 6. And, please, explain what software was used (I guess R).
8) In the text Supplementary Table S1 is the only one mentioned, the rest of the supplementary tables are not mentioned. And I guess that they should be reordered. In addition, Figures 1A, 1C and 1E are not the first to mention; Figure S4 is not mentioned. Please, check that every table and figure is mentioned and in order.
